# Measurement of Oral Moisture on Oral Dryness Patients

**DOI:** 10.3390/geriatrics5020028

**Published:** 2020-04-30

**Authors:** Fumi Mizuhashi, Kaoru Koide, Shuji Toya, Tomoko Nashida

**Affiliations:** 1Department of Removable Prosthodontics, The Nippon Dental University School of Life Dentistry at Niigata, Niigata 951-8580, Japan; koide@ngt.ndu.ac.jp; 2Dry Mouth Clinic, Oral and Maxillofacial Surgery, The Nippon Dental University Niigata Hospital, Niigata 951-8580, Japan; toya@ngt.ndu.ac.jp; 3Department of Biochemistry, The Nippon Dental University School of Life Dentistry at Niigata, Niigata 951-8580, Japan; nashida@ngt.ndu.ac.jp

**Keywords:** oral dryness, oral moisture, measuring pressure

## Abstract

Many elderly patients have oral dryness; thus, it is necessary to evaluate the oral moisture in a clinical setting. The aim of this study was to clarify the importance of controlling the measuring pressure of the oral moisture-checking device. The influence of the measuring pressure of the oral moisture-checking device was examined using agar under 10 measuring pressure conditions (Kruskal–Wallis test). Fifty-five oral dryness patients were examined the lingual moisture using the device with and without a tongue depressor. The tongue depressor was placed underneath the tongue to support it during the measurement. The mean value and the coefficient of variation of five measurements was evaluated (paired t-test or Wilcoxon signed-ranks test). The agar moisture values changed according to the measuring pressure (*p* < 0.05). The lingual moisture value with the tongue depressor was higher than that without the tongue depressor (*p* < 0.05). The coefficient of variation with the tongue depressor was smaller than that without the tongue depressor (*p* < 0.01). The results of this study indicated that the measuring pressure of oral moisture-checking device influenced the measurement value, and it is necessary to support the tongue for the measurement of lingual mucosal moisture in a uniform manner.

## 1. Introduction

A decrease in the amount of salivary secretion results in xerostomia [1], also termed oral dryness. This in turn can lead to other symptoms associated with the oral cavity and pharynx [2,3,4,5,6,7,8]. Many elderly patients with dental disorders have subjective oral dryness; thus, it is necessary to evaluate this situation in the clinical setting. An easy method with which to evaluate oral dryness without the influence of the individual’s oral function or overall condition is required [9,10].

The oral moisture-checking device Mucus^®^ (Life Co., Saitama, Japan) easily measures oral mucosal moisture within two seconds [11]. This device is the first device to measure the oral moisture, and was recently used to evaluate oral dryness among elderly populations residing in nursing homes, patients in the intensive care unit, and patients who have suffered strokes because the results are not influenced by oral function or overall condition [12]. Oral moisture has been measured at both the lingual and buccal mucosae with a measuring pressure of about 200 g/cm^2^ [11]. A previous study showed that stimulated saliva and oral moisture at the lingual mucosa were associated with subjective oral dryness, but no relationship between oral moisture at the buccal mucosa and subjective oral dryness was found [13]. Therefore, measurement of the oral moisture at the lingual mucosa was considered to be desirable. Lingual mucosal moisture is measured at the surface of the tongue by touching the sensor of the device and applying the measuring pressure on the tongue in clinical use. However, it is difficult to add the measuring pressure of the device on the tongue because the tongue sometimes moves downward with the application of measuring pressure. According to the manufacturer’s instructions, the measured value is not accurate when the measuring pressure is not 200 g/cm^2^.

The hypothesis of the present study was that the measurement value of the oral moisture-checking device would be influenced by the measuring pressure, and if a sufficient measuring pressure is applied; the lingual mucosal moisture will be measured more easily regardless of the muscle power of the tongue of the individuals. In this study, a tongue depressor was considered to place underneath the tongue to support it during the measurement, although, the tongue depressor has not been used clinically. The purpose of this study was to test the following null hypothesis: the measurement value of the oral moisture-checking device was not influenced by the measuring pressure.

## 2. Materials and Methods 

### 2.1. Evaluation of Influence of Measuring Pressure

The material used for this study was agar, and the examination was performed with reference to the study by Saito et al [14]. Agar powder (Wako Pure Chemical Industries, Ltd., Osaka, Japan) was mixed with distilled water and poured into a plastic dish (80 mm in diameter, 10 mm in height) to the height of 10 mm. Three agar concentrations were examined, 2%, 4%, and 6%. The agar was cooled at room temperature and then placed in a refrigerator overnight with a lid on the dish to avoid moisture evaporation.

An oral moisture-checking device (Mucus^®^, Life Co., Saitama, Japan) (Figure 1) was used for the measurements. The device was fixed to a holder at the point of contact between its sensor and the surface of the agar (Figure 2). It was then pushed to pressures of 75, 100, 125, 150, 175, 200, 225, 250, 275, and 300 g/cm^2^ using a push–pull gauge (PP-705-500^®^, Teclock Co., Nagano, Japan). The measurement was repeated three times. Statistical analysis (Kruskal–Wallis test and Scheffé method) was performed to analyze the differences in the measurement values according to the measuring pressure. A nonparametric test was selected for analysis because normality was not recognized in each group. Statistical analysis was performed using statistical analysis software (SPSS 17.0^®^, SPSS JAPAN, Tokyo, Japan), and differences of *α* < 0.05 were considered significant.

### 2.2. Measurement of Oral Moisture on Oral Dryness Patients

The participants of this study were 55 oral dryness patients (five men, 50 women; mean age, 70.8 ± 10.0 years). The sample size was determined by power analysis. The participants were the patients that came to the dry mouth clinic at The Nippon Dental University Niigata Hospital. All subjects gave their informed consent for inclusion before they participated in the study. The study was conducted in accordance with the Declaration of Helsinki, and the protocol was approved by the Ethics Committee of The Nippon Dental University School of Life Dentistry at Niigata (#49).

Oral moisture was measured using an oral moisture-checking device (Mucus^®^) at the lingual mucosa with and without a tongue depressor (Men-tip^®^, Nihon-Menbow Co., Tokyo, Japan). One operator trained in the ability to maintain a measuring pressure of 200 g/cm^2^ measured the oral moisture of all participants. The operator trained to add the measuring pressure 200 g/cm^2^ using a scale and the accuracy of the pressure was evaluated by a test adding the pressure of 200 g/cm^2^ without seeing the scale. For the measurement of lingual mucosal moisture without the tongue depressor, the participants were asked to stick out their tongue, and then the operator measured the lingual mucosal moisture by touching the sensor of the device on the surface of the tongue 10 mm from the apex linguae and applying the measuring pressure on the tongue (Figure 3a). For the measurement of lingual mucosal moisture with the tongue depressor, the operator positioned the tongue of the participants over the tongue depressor, and then, the lingual mucosal moisture was measured at the surface of the tongue 10 mm from the apex linguae by applying the measuring pressure on the tongue (Figure 3b). The tongue depressor was positioned under the tongue to support it during the measurement. These examinations were performed more than two hours after a meal to prevent any influence of the meal. Measurements were performed five times under each condition, and the median moisture value was used for analysis. The coefficient of variation of five measurements under each condition was calculated to evaluate the dispersion of the measurement. Statistical analysis was performed to compare the measurement values using Wilcoxon signed-ranks test because normality of the difference between the two groups was not recognized. Coefficient of variation was analyzed by paired *t*-test because normality of the difference between the two groups was recognized. Statistical analysis was performed using statistical analysis software (SPSS 17.0^®^), and differences of *α* < 0.05 were considered significant.

## 3. Results

### 3.1. Agar Moisture at Different Measuring Pressures

The influence of the measuring pressure on the measurement value was examined using agar plates. Figure 4 shows the mean value and standard deviation of the agar moisture at different measuring pressures. The measurement values showed broad variation under a measuring pressure of 200 g/cm^2^, but remained stable at pressures above this level. The results of multiple comparisons showed that statistically significant differences were observed between the measuring pressures ≤ 100 and ≥200 g/cm^2^ on the three agar concentrations (*p* < 0.05).

### 3.2. Lingual Mucosal Moisture

Table 1 shows the mean value and standard deviation of lingual mucosal moisture of oral dryness patients with and without tongue depressor. The measurement value with the tongue depressor was higher than that without the tongue depressor (*p* < 0.05). The coefficient of variation with the tongue depressor was smaller than that without the tongue depressor (*p* < 0.01).

## 4. Discussion

The oral moisture-checking device can be used independent of the individual’s oral function and overall condition. However, it is difficult to add the measuring pressure on the tongue. We investigated the influence of measuring pressure of the oral moisture-checking device on the measurement value to establish a uniform measurement manner for the evaluation of oral dryness.

The oral moisture-checking device used in this study was modified from a skin moisture-checking device to measure the moisture of the epithelium at a depth of several tens of micrometers within the area of a square centimeter [11]. According to the manufacturer’s instructions, the principle of the oral moisture-checking device is that the epithelial moisture is measured as capacitance. The dielectric constant of water is much higher than that of other substances; therefore, the percentage of water in the substance can be checked by measuring the dielectric constant of the substance. The moisture of the substance is measured by calculating the capacitance from the dielectric constant of the substance, and the reliability of the data was previously demonstrated by comparison with the dry weight method [11].

The relationship between the measuring pressure and the moisture value was investigated using agar plates. The measuring pressure influenced the moisture value of the agar at all concentrations, and the moisture values were stable at measuring pressures of ≥200 g/cm^2^, which has been reported as proper pressure. This result suggested that the measurement value would not be influenced by the measuring pressure when the measuring pressure was ≥200 g/cm^2^.

When the sensor of the device presses the tongue at a point 10 mm from the apex, the tongue easily moves downward according to the degree of the individual’s power to support his or her own tongue, and the measuring pressure would be decreased in the case of the power was weak. In individuals with oral dysfunction, such measurement may be difficult depending on their condition. The use of a tongue depressor was thought to be an advantage because of support of the tongue during the measurement. The average lingual mucosal moisture level with the tongue depressor was larger than that without the tongue depressor. The reason of this result could be considered from the result of the examination using agar as the sufficient measuring pressure was added on the tongue by using the tongue depressor, and the measurement value became larger when the tongue depressor was used. Though the measuring pressure added on the tongue was not clear, the measurement value would not be influenced by the measuring pressure when the measuring pressure was ≥200 g/cm^2^; then, the measurement value obtained using the tongue depressor could be considered more accurate than that obtained without the tongue depressor. The average coefficient of variation with the tongue depressor was smaller than that without the tongue depressor. That is, measurement value of the lingual mucosal moisture with the tongue depressor was stable. The stable measuring pressure would be added by using the tongue depressor, and then the stable measurement value of the lingual mucosal moisture could be obtained. The measurement value with the measuring pressure of 300 g/cm^2^ was more stable than that with the measuring pressure of 200 g/cm^2^. In the future study, the best measuring pressure of the oral moisture-checking device should be investigated on the participants.

These results indicate that it is necessary to consider the influence of the measuring pressure for the measurement of lingual mucosal moisture, and it is effective to support the tongue during the measurement of lingual mucosal moisture for obtaining the stable measuring pressure and the stable measurement value. The tongue should be supported during the measurement of lingual mucosal moisture in order to evaluate the oral moisture in a uniform manner.

## 5. Conclusions

The results of this study indicate that measuring pressure of oral moisture-checking device influenced the moisture value. The measurement of lingual mucosal moisture could be performed in a uniform manner by supporting the tongue during the measurement.

## Figures and Tables

**Figure 1 geriatrics-05-00028-f001:**
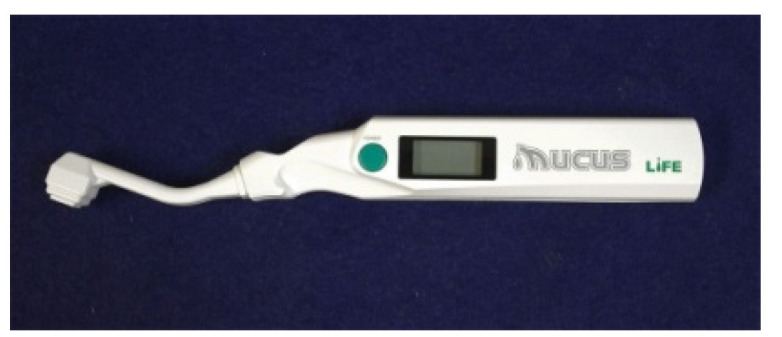
An oral moisture-checking device (Mucus^®^).

**Figure 2 geriatrics-05-00028-f002:**
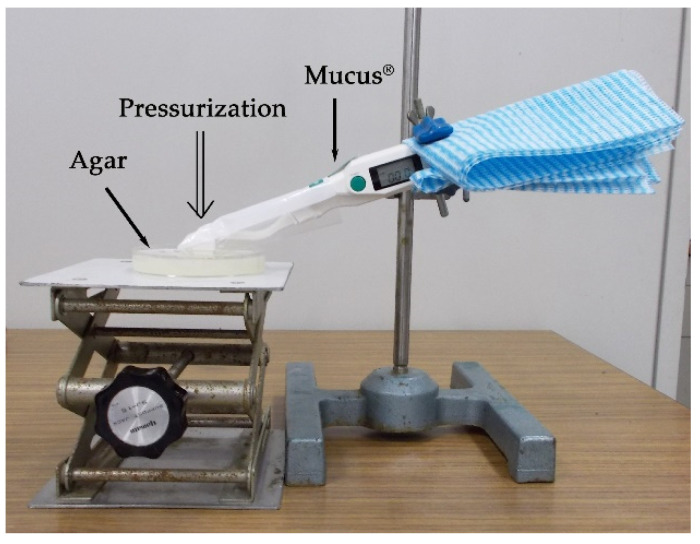
Measurement of agar moisture.

**Figure 3 geriatrics-05-00028-f003:**
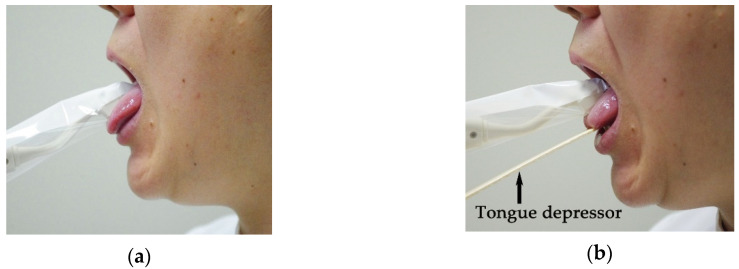
Measurement of oral moisture at lingual mucosa. (**a**) Without tongue depressor. Participants were asked to stick out their tongue. (**b**) With tongue depressor. Tongue depressor was positioned under tongue.

**Figure 4 geriatrics-05-00028-f004:**
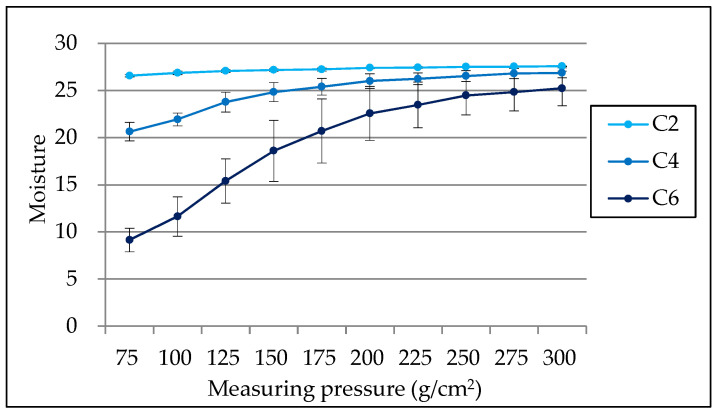
Mean value and standard deviation of agar moisture at each measuring pressure. C2: 2% agar, C4: 4% agar, C6: 6% agar.

**Table 1 geriatrics-05-00028-t001:** Mean value and standard deviation of lingual mucosal moisture with and without tongue depressor.

	With Tongue Depressor	Without Tongue Depressor	*p* Value
Measurement value	29.52 ± 2.33	29.05 ± 2.59	<0.05
Coefficient of variation	0.99 ± 0.80	2.02 ± 1.53	<0.01

mean values ± SD

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
