# Peer review of "Measurement of Oral Moisture on Oral Dryness Patients"

_geriatrics, 2020, doi:10.3390/geriatrics5020028_

Round 1

Reviewer 1 Report

It's not clear how the clinical measurement was standardized. Have to be better explain how the operator trained in the ability to maintain a measuring pressure of 200 g/cm2 and what kind of measurement was performed to evaluate this.

Reviewer 2 Report

In this study, Mizuhashi et al. clarified the importance of controlling the measuring pressure of oral moisture-checking device by Agar experiment and the test of lingual mucosal moisture in patients. And the authors claimed that that the measuring pressure of oral moisture-checking device influenced the measurement value, and it is necessary to support the tongue for the measurement of lingual mucosal moisture in a uniform manner. Its significance is clear, and the conclusions are well demonstrated. Thus, the authors provided an appropriate and accurate method for the evaluation of xerostomia, which provides a basis for the accurate clinical evaluation of oral moisture. This which makes the study suitable for this Journal. Several minor concerns listed below.

CONCERNS
1. Is this device already in clinical use? What is the clinical procedure (with or without the tongue depressor)?

2. Are there any other ways to measure oral moisture? How does it compare?
3. It seems that 300 g/cm2 is more accurate than 200 g/cm2, why not compare the pressure changes when measuring patients?
